# New 8-*C*-*p*-Hydroxylbenzylflavonol Glycosides from Pumpkin (*Cucurbita moschata* Duch.) Tendril and Their Osteoclast Differentiation Inhibitory Activities

**DOI:** 10.3390/molecules25092077

**Published:** 2020-04-29

**Authors:** Kiok Kim, Joo-Hee Choi, Jisu Oh, Ji-Yeon Park, Young-Min Kim, Jae-Hak Moon, Jong-Hwan Park, Jeong-Yong Cho

**Affiliations:** 1Department of Food Science & Technology, Chonnnam National University, Gwangju 61186, Korea; kiwi6287@naver.com (K.K.); dhwltn7117@naver.com (J.O.); u9897854@jnu.ac.kr (Y.-M.K.); nutrmoon@jnu.ac.kr (J.-H.M.); 2Laboratory of Animal Medicine, College of Veterinary Medicine and BK21 Plus Project team, Chonnam National University, Gwangju 61186, Korea; cjh522@hanmail.net (J.-H.C.); smileonjy@hanmail.net (J.-Y.P.); 3Laboratory Animal Center, Daegu-Gyeongbuk Medical Innovation Foundation, Daegu 41061, Korea

**Keywords:** pumpkin tendril, *Cucurbita moschata* Duch., 8-*C*-*p*-hydroxybenzylflavonol, 8-*C*-*p*-hydroxybenzylflavonol glycosides, osteoclast differentiation inhibition

## Abstract

Six new 8-*C*-*p*-hydroxybenzylflavonol glycosides were isolated from a hot water extract of pumpkin (*Cucurbita moschata* Duch.) tendril and elucidated as 8-*C*-*p*-hydroxybenzylquercetin 3-*O*-rutinoside, 8-*C*-*p*-hydroxybenzoylquercetin 3-*O*-β-D-glucopyranoside, 8-*C*-*p*-hydroxybenzylkaempferol 3-*O*-(α-L-rhamnopyranosyl(1→6)-β-D-galactopyranoside, 8-*C*-*p*-hydroxybenzoylkaempferol 3-*O*-rutinoside, 8-*C*-*p*-hydroxybenzylisorhamnetin 3-*O*-rutinoside, and 8-*C*-*p*-hydroxybenzylisorhamnetin 3-*O*-(α-L-rhamnopyranosyl(1→6)-β-D-galactopyranoside. Their chemical structures were determined using nuclear magnetic resonance (NMR) and electrospray ionization-mass spectrometer (ESIMS) analyses. The 8-*C*-*p*-hydroxybenzylflavonol glycosides were found to inhibit the receptor activator of nuclear factor-κB (RANKL)-induced osteoclast differentiation of bone marrow derived macrophage (BMDM), an osteoclast progenitor. Additionally, 8-*C*-*p*-hydroxybenzylflavonol glycosides effectively reduced the expression of osteoclast-related genes, such as tartrate-resistant acid phosphatase, cathepsin K, nuclear factor activated T-cell cytoplasmic 1, and dendritic cell specific transmembrane protein in RANKL-treated BMDMs. These results indicate that the 8-*C*-*p*-hydroxybenzylflavonol glycosides may be the main components responsible for the osteoclast differentiation inhibitory effect of pumpkin tendril.

## 1. Introduction

Bone remodeling by osteoblasts, which regulate bone formation, and osteoclasts, which regulate bone resorption, is essential for maintaining bone mass. Disruption of the bone formation/resorption balance due to abnormal osteoclast activity leads to skeletal diseases, such as osteoporosis, rheumatoid arthritis, and periodontal disease [1]. Osteoclasts are multinucleated cells differentiated from hematopoietic precursor cells of monocyte/macrophage lineage in the bone marrow. They are characterized by their specialized membrane structures, clear zones and ruffled borders, acid secretion, and lytic enzymes that degrade mineralized bone matrices [2,3]. Osteoclast differentiation and maturation require two critical cytokines, the receptor activator of nuclear factor-kappa B (NF-κB) ligand (RANKL) and macrophage colony-stimulating factor (M-CSF) [4]. The binding of RANKL to its receptor activator of NF-κB (RANK) leads to the activation of key transcription factors such as nuclear factor activated T-cell cytoplasmic 1 (NFATc1) [5,6], which enhances osteoclast formation by increasing the expression of osteoclast-related genes [7]. Thus, inhibiting osteoclast differentiation and activation is a key therapeutic strategy for treating bone metabolic diseases.

Pumpkin, a gourd-like squash of the genus *Cucurbita* and the family *Cucurbitaceae*, is a nutritious vegetable that is widely consumed throughout the world. The nutritional and health benefits of pumpkin seeds and fruits have received considerable attention in recent years. Studies have demonstrated that *Cucurbita moschata* has extensive bioactivities, including hepatoprotective [8], anti-diabetes [9], anti-cancer [10], and anti-obesity properties [11]. In addition, the fruit has been reported to attenuate fatigue and enhance exercise performance in mice [12].

Pumpkin byproducts, such as pumpkin leaves, stems, and tendrils, have been used as vegetables and in medical materials in Asia. In particular, blanched or steamed pumpkin leaves are widely consumed in Korea. A water-soluble extract of the pumpkin stem has been reported to exert anti-obesity effects via control of lipid metabolism in mice with obesity induced by a high-fat diet [11]. They recently reported that dehydrodicaniferyl alcohol isolated from pumpkin stems inhibited adipogenic and lipogenic effects in 3T3-cells and primary mouse embryonic fibroblasts [13] and prevented ovariectomy-induced bone loss by inhibiting osteoclast differentiation [14,15]. Pumpkin tendril has also been found to inhibit inflammation, with rutin being identified as an anti-inflammatory compound in this material [16]. Recently, we reported that the water extract of pumpkin tendrils suppressed RANKL-induced osteoclast differentiation via downregulation of p38 and extracellular-signal-regulated kinase (ERK) [17]. Nevertheless, the results of our series of studies on pumpkin tendril suggested the presence of other unidentified bioactive compounds. Therefore, we have isolated and identified the chemical constituents from the water extract of pumpkin tendril.

In this study, we describe the isolation and structural elucidation of six new 8-*C*-*p*-hydroxybenzylflavonol glycosides from pumpkin tendril. In addition, the osteoclast differentiation inhibitory activities of the isolated compounds were evaluated. 

## 2. Results and Discussion

### 2.1. Structural Elucidation of the Isolated Compounds

Compound **1** was obtained as a yellow amorphous powder. The molecular formula of **1** was determined to be C_34_H_36_O_17_ (MW 716) by high resolution electrospray ionization source (ESIMS; HRESIMS, negative, *m/z* 715.1873 [M − H]^−^). The ^1^H nuclear magnetic resonance (NMR) spectrum (Appendix A) of **1** exhibited the presence of the quercetin skeleton, including three tri-substituted benzene ring proton signals of the B ring at δ 7.72 (1H, d, *J* = 2.1 Hz, H-2′), 6.84 (1H, d, *J* = 9.0 Hz, H-5′), and 7.46 (1H, dd, *J* = 9.0, 2.1 Hz, H-6′), and a methine proton signal of the A ring at δ 6.31 (1H, s, H-6). Additionally, the proton signals related to the *p*-hydroxybenzyl moiety were observed at δ 4.05 (2H, s, H-1′′), 7.07 (2H, d, *J* = 8.4 Hz, H-3′′ and H-7′′), and 6.64 (2H, d, *J* = 8.4 Hz, H-4′′ and H-6′′). Moreover, the ^1^H NMR spectrum indicated the presence of β-D-glucopyranose and α-L-rhamnopyranose moieties, including two anomeric proton signals at δ 5.08 (1H, d, *J* = 7.8 Hz, H-1′′′) and 4.52 (1H, d, *J* = 1.2 Hz, H-1′′′′), a methyl proton signal at δ 1.12 (3H, d, *J* = 6.6 Hz, H-6′′′′), and other proton signals at δ 3.87−3.18, which were assigned based on their coupling constants and the correlations observed in the homonuclear correlation spectroscopy (^1^H−^1^H COSY) spectrum (bold lines). These results were also supported by the presence of 34 carbon signals assignable to quercetin and *p*-hydroxybenzyl moieties, including a carbonyl carbon at δ 179.8 (C-4), in the ^13^C NMR spectrum (Appendix A). Based on the MS and 1D NMR results, **1** was proposed to be quercetin coupled with a *p*-hydroxybenzyl group and two sugars. Their partial structures were assigned as quercetin, 1-methyl-*p*-hydroxybenzene, and rutinose based on HSQC, ^1^H-^1^H COSY, and heteronuclear multiple bond correlation (HMBC) spectra. In the HMBC spectrum, δ 5.08 (H-1′′′) correlated with δ 135.8 (C-3) (Figure 1, arrows), indicating that the rutinose moiety connected to the C-3 of quercetin via an ether linkage. In particular, the proton signal at δ 4.05 (H-1′′) was observed to correlate with the carbon signals at δ 163.9 (C-7) and 156.0 (C-9) in the HMBC spectrum (Figure 1, arrows), confirming that the *p*-hydroxybenzyl moiety was coupled to the A ring of quercetin 3-*O*-rutinose. The exact structure of **1** was determined from a nuclear Overhauser effect (NOE) experiment in DMSO-*d*_6_. In particular, irradiation at δ 4.05 (H-1′′) led to enhancements of the signals at δ 7.72 (H-2′) and 7.46 (H-6′), indicating that the *p*-hydroxybenzyl moiety was positioned at C-8 of the quercetin 3-*O*-rutinose moiety. Therefore, compound **1** was determined to be 8-*C*-*p*-hydroxybenzylquercetin 3-*O*-rutinoside (Figure 1). 

The ^1^H and ^13^C NMR spectra of **2**–**6** were similar to those of **1** (Appendix A). The 8-*C*-*p*-hydroxybenzylflavonol moiety was identical, while the number of hydroxyl and methoxy groups and their positions on the *p*-hydroxybenzylflavonol structure, and types of sugars were differed, suggesting that they were 8-*C*-*p*-hydroxybenzylflavonol glycosides. The structures of **2**–**6** were determined using 2D-NMR (HSQC, HMBC, ^1^H–^1^H COSY, and NOE) and MS data. Consequently, these compounds were elucidated as the 8-*C*-*p*-hydroxybenzylflavonol glycosides shown in Figure 1.

The molecular formula of **2** was determined to be C_28_H_26_O_13_ (MW 570) by negative HRESIMS analysis (*m/z* 569.1296 [M − H]^−^). The ^1^H and ^13^C NMR spectra of **2** were very similar to those of **1** (Appendix A). However, the proton and carbon signals corresponding to the rhamnose moiety were not observed in the ^1^H and ^13^C NMR spectra of **2**. The sugar moiety was assigned as β-D-glucopyranose based on the presence of an anomeric proton signal at δ 5.25 (1H, d, *J* = 7.5 Hz, H-1′′′), other sugar proton signals at δ 3.22–3.71 and their coupling constant values (*J* > 8.5 Hz), and the proton–proton correlations observed in the ^1^H−^1^H COSY spectrum ((Figure 1, bold lines). The HMBC correlation observed between δ 5.27 (H-1′′′) and δ 134.8 (C-3) (Figure 1, arrows) indicated that β-D-glucopyranose was connected to *C*-3 of quercetin via an ether linkage. Therefore, compound **2** was determined to be 8-*C*-*p*-hydroxybenzylquercetin 3-*O*-β-d-glucopyranoside (Figure 1).

The molecular formula of **3** was established to be C_34_H_36_O_16_ (MW 700) by negative HRESIMS analysis (*m/z* 699.1927 [M − H]^−^). Comparison of its ^1^H NMR spectrum with that of **1** suggested a partial structure consisting of kaempferol and galactose rather than quercetin and glucose. Specifically, the presence of the kaempferol moiety was suggested by the proton signals corresponding to a *p*-substituted benzene ring at δ 7.94 (2H, d, *J* = 9.0 Hz, H-3′ and H-5′) and 6.84 (2H, d, *J* = 9.0 Hz, H-2′ and H-6′) in the ^1^H NMR spectrum (Appendix A). The presence of the galactose moiety was inferred based on the anomeric proton signal at δ 5.02 (1H, d, *J* = 7.5 Hz, H-1′′′), other proton signals at δ 3.79–3.36, and the coupling constants (*J* = 2.5 and 3.5 Hz) of its H-4 at δ 3.76 (Appendix A). The MS and ^1^H NMR results suggested that **3** was 8-*C*-*p*-hydroxybenzylkaempferol rhamnopyranosylgalactopyranoside; its ^13^C NMR spectrum and 2D-NMR spectra (Appendix A and Figure 1) supported this conclusion. HMBC correlations (Figure 1, arrows) were observed between δ 3.74 (H-6′′′) and δ 100.4 (C-1′′′′), δ 4.52 (H-1′′′′) and δ 65.8 (C-6′′′), and δ 5.02 (H-1′′′) and δ 134.3 (C-3). Therefore, compound **3** was determined to be 8-*C*-*p*-hydroxybenzylkaempferol 3-*O*-(α-L-rhamnopyranosyl (1→6)-β-D-galactopyranoside (Figure 1).

The molecular formula of **4** was established to be C_34_H_36_O_16_ (MW 700) from negative HRESIMS analysis (*m/z* 699.1927 [M − H]^−^). When the ^1^H NMR spectra of **1** and **3** were compared to that of **4**, partial structures assignable to kaempferol and rutinose were observed for **4**; the presence of such structures was confirmed from its ^13^C NMR spectrum and 2D-NMR spectra. In particular, HMBC correlations (Figure 1, arrows) were observed between δ 3.80 (H-6′′′) and δ 103.4 (C-1′′′′), δ 4.52 (H-1′′′′) and δ 65.8 (C-6′′′), and δ 5.12 (H-1′′′) and δ 134.1 (C-3). On this basis, compound **4** was determined to be 8-*C*-*p*-hydroxybenzylkaempferol 3-*O*-rutinoside (Figure 1).

In the ^1^H and ^13^C NMR spectra of **5**, proton and carbon signals corresponding to a methoxy group were observed at δ 3.84 and δ 55.3, while all other proton signals were very similar to those of **1** (Appendix A). The HMBC correlation (Figure 1, arrows) between δ 3.82 (-OCH_3_) and δ 146.9 (C-3′) indicated that the methyl group was connected to the C-3 of quercetin via an ether linkage. Consequently, **5** was suggested to be 8-*C*-*p*-hydroxybenzylisorhamnetin 3-*O*-rutinoside based on the ^13^C NMR and 2D NMR experiments (Appendix A). However, the ^1^H NMR spectrum suggested the presence of another minor compound (**6**) in addition to the main compound (**5**). Most of the proton signals of the minor compound (**6**) overlapped with those of **5**, but the chemical shift values of the protons from the flavonol B ring and the methoxy group of **6** were significantly different from those of **5** (Appendix A). The ratio of the main compound (**5**) to the minor compound (**6**) was determined to be approximately 6:4 from the intensities of the flavonol B ring and methoxy proton signals in the ^1^H NMR spectrum. The sugar carbon signals in the ^13^C NMR spectrum of **6** were very similar to those of **3**. In particular, the presence of the galactose in **6** was assigned from the coupling constants (*J* = 2.5 and 3.5 Hz) of its H-4 at δ 3.75 in the ^1^H NMR experiment (Appendix A). Consequently, **6** was suggested to be 8-*C*-*p*-hydroxybenzylisorhamnetin 3-*O*-rhamnopyranosylgalactopyranoside based on its 2D NMR spectra (Figure 1). Negative HRESIMS analysis (*m/z* 729.2036 [M − H]^−^) established that the molecular formula of both **5** and **6** was C_35_H_38_O_17_ (MW 730). Therefore, compounds **5** and **6** were determined to be 8-*C*-*p*-hydroxybenzylisorhamnetin 3-*O*-rutinoside and 8-*C*-*p*-hydroxybenzylisorhamnetin 3-*O*-(α-L-rhamnopyranosyl(1→6)-β-D-galactopyranoside, respectively (Figure 1).

8-*C*-*p*-Hydroxybenzylflavonol derivatives, such as 8-*C*-*p*-hydroxybenzylapigenin, 8-*C*-*p*-hydroxybenzylluteolin, 8-*C*-*p*-hydroxybenzyldiosmetin, 8-*C*-*p*-hydroxybenzylquercetin, and 8-*C*-*p*-hydroxybenzylkaempferol, have previously been found in other plants [18,19]. However, to the best of our knowledge, the 8-*C*-*p*-hydroxybenzylflavonol glycosides **1**–**6** isolated from pumpkin tendril represent new compounds.

### 2.2. Inhibition of RANKL-Induced Osteoclast Differentiation in Bone Marrow Derived Macrophages (BMDMs) by the Isolated Compounds

Bone marrow derived macrophages are osteoclast precursors that differentiate into osteoclasts when stimulated with RANKL [20]. RANKL-induced osteoclast differentiation was measured using tartrate-resistant acid phosphatase (TRAP) stain, an osteoclast marker protein [21]. To determine the suitable concentration of new compounds on osteoclast differentiation, cell viability was assessed with **1**, representatively. Bone marrow-derived macrophages (BMDMs) were pretreated with various concentrations (12.5, 25, 50, 75 μM) of **1** in the presence of M-CSF for 24 h and cell viability was determined by 3-(4,5-dimethylthiazol-2-yl)-2,5-diphenyltetrazolium bromide (MTT). Compound **1** showed significant cytotoxicity at a concentration of 75 μM, whereas there was no cytotoxicity under 50 μM (Appendix A). However, a concentration of 40 μM was used for **5**+**6** purified as a mixture in this study, as cytotoxicity was observed at a concentration of 50 μM during osteoclast differentiation (Appendix A). Therefore, we determined the concentrations of 50 μM (**1**–**4**) and 40 μM (**5**+**6** mixture) to evaluate the inhibitory effect of RANKL-induced osteoclast differentiation. The results showed that treatment with RANKL effectively induced osteoclast differentiation of BMDMs, and that the addition of compounds **1**–**5**+**6** decreased the size and number of osteoclasts (Figure 2A,B). To investigate whether the suppression of osteoclast differentiation by these compounds was due to the potential toxicity of these products, we performed cell viability assays in the culture after 24 h. The results showed that the concentration of compounds used in the osteoclast differentiation experiments did not affect the cell viability (Figure 2C).

Upon binding to RANK on the surface of osteoclast precursor, RANKL induces the expression of NFATc1, a master regulator of osteoclastogenesis that regulates osteoclast-related genes such as TRAP, Cathepsin K, and DC-STAMP [22,23,24]. TRAP and Cathepsin K are involved in the degradation of bone. DC-STAMP is required for cell-to-cell fusion for the generation of intact multinucleated osteoclasts. Thus, we examined whether the isolated compounds inhibited RANKL-induced gene expression. Real-time PCR analysis showed that the mRNA expression levels of NFATc1, TRAP, Cathepsin K, and DC-STAMP increased significantly due to the stimulation of RANKL treatment (Figure 3A–C), but were effectively downregulated after treatment with **1**–**4** (Figure 3A–C). Treatment of the **5**+**6** mixture trended towards inhibiting the expression of the RANKL-induced NFATc1, TRAP, and Cathepsin K mRNA, although this was not statistically significant (Figure 3A–C). However, the expression of the DC-STAMP mRNA was effectively attenuated by all of the tested compounds, including the **5**+**6** mixture (Figure 3D). It can be seen that the **5**+**6** mixture reduced the size and proliferation of mature osteoclasts as a result of Figure 2A,B by suppressing DC-STAMP, which plays a role in cell fusion.

Mature osteoclasts caused the formation of pits in the presence of M-CSF and RANKL. Therefore, we investigated whether each compound has the potential to inhibit the function of osteoclastic bone resorption. In order to examine the effect of isolated compounds on osteoclastic bone resorption, BMDMs were cultured on an Osteo assay surface plate and pretreated with **1**–**4** (50 μM) or **5**+**6** mixture (40 μM) in the presence of M-CSF (25 ng/mL) for 2 h, and subsequently stimulated with RANKL (100 ng/mL) for 4 days. Consistent with the size and number of osteoclasts (Figure 2A,B), the isolated compounds remarkably reduced the formation of bone resorption by inhibiting the area and number of pits on the Osteo surface (Figure 4A,B). These findings suggest that six new 8-*C*-*p*-hydroxybenzylflavonol glycoside (**1–6**) compounds, especially compounds **2** and **3**, isolated from the hot water extract of pumpkin tendril, inhibit osteoclast differentiation by decreasing the expression of genes related to osteoclastogenesis. Furthermore, we confirmed that bone resorption was inhibited by isolated compounds. Recently, we reported that a water extract of pumpkin tendril suppresses RANKL-induced osteoclast differentiation and bone resorption by suppressing RANKL-mediated p38 and ERK phosphorylation [17]. In addition, water extract of pumpkin tendril treatment decreased the expression of osteoclast-related genes such as NFATc1, TRAP, Cathepsin K and TRAP [17]. In our study, new compounds isolated from a water extract of pumpkin tendril showed different levels of inhibitory effects at the same concentration. This means that the inhibitory effect of pumpkin tendril extract on osteoclastogenesis acted through a complex mechanism of several compounds. However, the detailed mechanism by which compound **5**+**6** suppresses RANKL-induced osteoclast differentiation and bone resorption remains to be elucidated.

### 2.3. Inhibitory Effect of Isolated Compounds on Production of Reactive Oxygen Species (ROS) during Osteoclast Differentiation

Evidence has gradually accumulated that RANKL stimulates intracellular reactive oxygen species (ROS) generation, which mediates RANKL-induced osteoclast differentiation [25,26,27]. Previous studies have reported that dietary flavonoids may help prevent osteoclast differentiation and bone loss by inhibiting ROS generation [28,29]. Thus, to assess the antioxidant effects of isolated compounds on the production of ROS during osteoclast differentiation, we detected the influence of compounds in RANKL-stimulated intracellular ROS accumulation using 2′, 7′-dichlorofluorescin diacetate (DCFDA). DCFDA is oxidized by ROS to the fluorescent dye DCF. The results showed that fluorescence of the isolated compound treatment groups were significantly attenuated to the levels of intracellular ROS generation compared with the RANKL treatment group during osteoclast differentiation (Figure 5A,B). These data suggested that the isolated compounds from pumpkin tendril could effectively inhibit RANKL-induced osteoclast differentiation by blocking intracellular ROS production. However, the isolated compounds from pumpkin tendril mediated osteoclast inhibition need further molecular studies.

## 3. Experimental Section

### 3.1. General Experimental Procedures

Nuclear magnetic resonance (NMR) spectra were obtained with ^unity^INOVA 600 and 500 spectrometers (Varian, Walnut Creek, CA, USA; KBSI Gwangju Center). The deuterated methanol (CD_3_OD) and deuterated dimethyl sulfoxide (DMSO-*d*_6_) used as NMR solvents were obtained from Merck Co. (Darmstadt, Germany). All mass spectra were acquired using a hybrid ion-trap time-of-flight mass spectrometer (SYNAPT G2, Waters, Cambridge, UK) equipped with an electrospray ionization source (ESIMS). Thin-layer chromatography (TLC) was carried out on silica gel TLC plates (silica gel 60 F254, 0.25 mm thickness, Merck, Darmstadt, Germany) and developed using *n*-butanol/acetic acid/H_2_O (4:1:1, *v*/*v*/*v*). The fractions were detected by spraying a 1% cerium IV sulfate ethanol solution onto the plate. An Amberlite XAD-2 column (Supelco, Bellefonte, PA, USA) and ODS gel (300 g, 40–63 μm, Merck, Billerica, MA, USA) were used for column chromatography. High performance liquid chromatography (HPLC, Shimadzu, Kyoto, Japan) for the purification and isolation of compounds from the fractions obtained using column chromatography was performed using a Spherisorb S5 ODS2 column (10 mm × 250 mm, Waters). The flow rate was 5.0 mL/min, and the eluents were monitored at 220 nm.

### 3.2. Materials and Chemicals

Dried pumpkin tendril (*Cucurbita Moschata* Duch.) was purchased at the Jjanggu-oppa agricultural market (Jinju, Gyeongsangnam-do, Korea). A voucher sample has been deposited in the warm-temperate forest arboretum located in Bogil Island, at Chonnam National University on Bogil Island.

### 3.3. Extraction and Partitioning

Powered pumpkin tendril (300 g) was extracted using hot water (9 L) at 95 ℃ for 15 min and filtered through cotton gauze. The residue was re-extracted using hot water (6 L). The hot water extract solutions were combined and partitioned using ethyl acetate (EtOAc, 18 L, three times) and water-saturated *n*-butanol (BuOH, 12 L, three times), successively. These fractions were concentrated in vacuo at 40 °C.

### 3.4. Isolation of the EtOAc Fraction

The EtOAc fraction (2.27 g) was chromatographed using an Amberlite XAD-2 column (3 × 60 cm) and eluted using a step-wise H_2_O/acetonitrile (MeCN) system with steps of 8:2, 6:4, 4:6, 2:8, and 0:10 (*v*/*v*, each step 800 mL). The fractions were subjected to TLC analysis and grouped into eight fractions (PT-A–PT-H). Fraction PT-E (599.1 mg) was chromatographed using an ODS column (2 × 55 cm) eluted with H_2_O/MeCN = 75:25, 70:30, 65:35, and 60:40 (*v*/*v*, each step 250 mL) to obtain eleven subfractions (PT-E1–PT-E11). Subfraction PT-E5 (31.0 mg) was purified using an ODS-HPLC system (isocratic elution, 20% MeCN) to give **1** (*t_R_* 24.1 min, 15.6 mg). Additionally, subfraction PT-E6 (128.9 mg) was purified using an ODS-HPLC system (isocratic elution, 23% MeCN) to give **2** (*t_R_* 12.7 min, 3.9 mg), **3** (*t_R_* 13.4 min, 7.6 mg), **4** (*t_R_* 14.5 min, 8.7 mg), and a mixture of **5** and **6** (*t_R_* 15.7 min, 17.1 mg).

#### 3.4.1. 8-*C*-*p*-Hydroxybenzylquercetin 3-*O*-rutinoside (**1**)

A yellow amorphous powder; UV CH_3_OH λ_max_ (nm) (log ε) 365 (4.32), 268 (4.48), 225 (4.49); ^1^H and ^13^C NMR data are listed in Appendix A; HRESIMS (negative) *m/z* 715.1879 [M − H]^−^ (calculated for C_34_H_35_O_17_, *m/z* 715.1874, +0.5 mDa).

#### 3.4.2. 8-*C*-*p*-Hydroxybenzoylquercetin 3-*O*-β-D-glucopyranoside (**2**)

A yellow amorphous powder; UV CH_3_OH λ_max_ (nm) (log ε) 364 (3.93), 269 (4.15), 209 (4.32); ^1^H and ^13^C NMR data are listed in Appendix A; HRESIMS (negative) *m/z* 569.1303 [M − H]^−^ (calculated for C_28_H_25_O_13_, *m/z* 569.1295, +0.8 mDa).

#### 3.4.3. 8-*C*-*p*-Hydroxybenzylkaempferol 3-*O*-(α-L-rhamnopyranosyl(1→6)-β-D-galactopyranoside (**3**)

A yellow amorphous powder; UV CH_3_OH λ_max_ (nm) (log ε) 355 (3.90), 270 (4.15), 225 (4.21); ^1^H and ^13^C NMR data are listed in Appendix A of Appendix A; HRESIMS (negative) *m/z* 699.1921 [M − H]^−^ (calculated for C_34_H_36_O_16_, *m/z* 699.1925, –0.4 mDa).

#### 3.4.4. 8-*C*-*p*-Hydroxybenzoylkaempferol 3-*O*-rutinoside (**4**)

A yellow amorphous powder; UV CH_3_OH λ_max_ (nm) (log ε) 356 (3.91), 270 (4.15), 225 (4.21); ^1^H and ^13^C NMR data are listed in Appendix A; HRESIMS (negative) *m/z* 699.1918 [M − H]^−^ (calculated for C_34_H_36_O_16_, *m/z* 699.1925, −0.7 mDa).

#### 3.4.5. Mixture of 8-*C*-*p*-hydroxybenzylisorhamnetin 3-*O*-rutinoside (**5**) and 8-*C*-*p*-hydroxybenzyl isorhamnetin 3-*O*-(α-L-rhamnopyranosyl(1→6)-β-D-galactopyranoside (**6**)

A yellow amorphous powder; UV CH_3_OH λ_max_ (nm) (log ε) 360 (4.14), 270 (4.33), 259 (4.32); ^1^H and ^13^C NMR data are listed in Appendix A; HRESIMS (negative) *m/z* 729.2036 [M − H]^−^ (calculated for C_35_H_37_O_17_, *m/z* 729.2031, +0.5 mDa).

### 3.5. Determination of the Osteoclast Differentiation Inhibitory Activity of the Isolated Compounds

#### 3.5.1. Osteoclast Differentiation and TRAP Staining

To generate osteoclasts, bone marrow derived macrophages (BMDMs) were prepared as described previously [30]. Briefly, mouse bone marrow cells were isolated from the femurs of mice between six and eight weeks old (Koatech, Gyunggi-Do, Korea). After lysing the red blood cells, the cells were incubated for 3 d at 37 °C in 5% CO_2_ in *α*-minimal essential medium (MEM) (Gibco) supplemented with 10% fetal bovine serum (FBS) (Gibco) and 1% penicillin/streptomycin in the presence of M-CSF (25 ng/mL). The cells that adhered to the bottom of the culture dish were classified as BMDMs. For osteoclast differentiation, the BMDMs were cultured in the presence of M-CSF (25 ng/mL) and RANKL (100 ng/mL) for an additional 4 days.

Osteoclast formation was determined by quantifying the cells that were positively stained by tartrate-resistant acid phosphatase (TRAP), which is an osteoclast enzyme marker. At the end of the differentiation, the cells were fixed with 3.7% formalin for 10 min and TRAP staining was performed using a commercial kit (Kamiya Biomedical, WA, USA) according to the manufacturer’s instructions. The number of TRAP-positive multinucleated cells (MNC; containing more than three nuclei) was counted under a light microscope.

#### 3.5.2. Cell Viability Assay

The effect of the isolated compounds **1**–**4** and **5**+**6** on the cell viability of the BMDMs was determined using a 3-(4,5-dimethylthiazol-2-yl)-2,5-diphenyltetrazolium bromide (MTT) assay [31]. Briefly, the BMDMs (4 × 10^5^ cells/mL) were seeded onto a 96-well plate and incubated with the isolated compounds in the presence of M-CSF (25 ng/mL). After 24 h, MTT (0.4 mg/mL) was added to each well, and the cells were incubated for a further 3 h. At the end of the incubation period, the insoluble formazan products were dissolved in dimethyl sulfoxide (DMSO), and the absorbance of each well at 570 nm was measured using a microplate reader.

#### 3.5.3. Real-Time Quantitative Polymerase Chain Reaction (qPCR)

BMDMs were seeded into 12-well plates, cultured in complete α-MEM containing M-CSF (25 ng/mL) and RANKL (100 ng/mL) and pretreated with the isolated compounds (40 or 50 μM) for 4 days. The total RNA was extracted from the cells using easy-BLUE (iNtRON, Gyunggi, Korea). Subsequently, 1 μg of the RNA was reverse transcribed to cDNA using an RT Premix reverse transcription system (Toyobo, Kita-ku, Japan) with oligo-dT18 primers. qPCR was then performed using the cDNA as a template with Power SYBR Green (Pcrbio, London, United Kingdom). β-Actin was used for normalization. Forty cycles of qPCR were performed at a temperature of 95 °C for 10 s and 60 °C for 60 s using the Rotor-Gene Q real-time PCR system (Qiagen, Hilden, Germany). The following primers were used: TRAP, 5′-CTGGAGTGCACGATGCCAGCGACA-3′ and 5′-TCCGTGCTCGGCGATGGACCAGA-3′; DC-STAMP, 5′-CCAAGGAGTCG TCCATGATT-3′ and 5′-GGCTGCTTTGATCGTTTCTC-3′; Cathepsin K, 5′-GGCCAA CTCAAGAAGAAAAC-3′ and 5′-GTGCTTGCTTCCCTTCTGG-3′; NFATc1, 5′-CTCGAAAGACAGTGGAGCAT-3′ and 5′-CGGCTGCCTTCCGTCTCATAG-3′; and *β*-actin, 5′-AGGCCCAGAGCAAGAGAG-3′ and 5′-TCAACATGATCTGGGTCATC-3′.

#### 3.5.4. Bone Pit Formation Assay

To observe the pit for bone resorption formation [32], BMDMs were seeded in complete α-MEM containing M-CSF (25 ng/mL) on an Osteo assay surface 24-well plate (Corning Inc., NY) and stimulated with RANKL (100 ng/mL) in the presence of the isolated compounds (40 or 50 μM) or vehicle (Veh). After 4 days, the media was aspirated from the wells and 10% sodium dodecyl sulfate (SDS) was added for 15 min to remove completely the cells. The wells were washed 3 times with distilled water, then the resorption area was observed under a light microscope and quantified using Image J software.

#### 3.5.5. Intracellular ROS Detection 

BMDMs were stimulated with RANKL without or with compounds for 48 h. Cells were washed twice with PBS, and then incubated with 2′, 7′-dichlorofluorescin diacetate (DCFDA, Sigma) for 20 min at 37 °C. The oxidative conversion of DCFDA to fluorescent DCF by ROS was detected and counted using a fluorescence microscope.

#### 3.5.6. Statistical Analysis

All statistical analyses were performed using GraphPad Prism version 5.01 (GraphPad, La Jolla, CA, USA). The statistical values were calculated using Student’s t-test and presented as mean ± standard deviation (SD). A result was considered statistically significant if the *p*-value was less than 0.05. 

## 4. Conclusions

Six new 8-*C*-*p*-hydroxybenzylflavonol glycosides (**1**–**6**) were isolated from a hot water extract of pumpkin tendril. These 8-*C*-*p*-hydroxybenzylflavonol glycosides inhibited osteoclast differentiation via suppression of the expression of osteoclast-related genes (TRAP, Cathepsin K, NFATc1, and DC-STAMP) in the RANKL-treated BMDMs. Pumpkin tendril can be regarded as a promising health-promoting food and source of medicinal compounds based on the high osteoclast differentiation inhibitory activity of the 8-*C*-*p*-hydroxybenzylflavonol glycosides found in the plant. The extent to which these compounds contribute to the anti-osteoclast effects of pumpkin tendril in vivo and how they are absorbed in the body remain to be investigated in future studies.

## Figures and Tables

**Figure 1 molecules-25-02077-f001:**
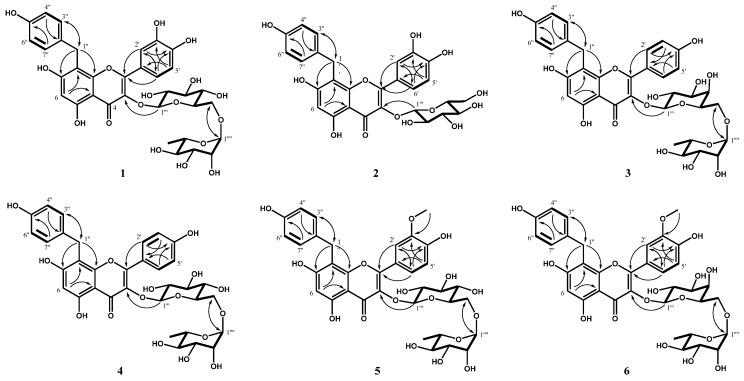
Structure of compounds **1**–**6** isolated from pumpkin tendril and their important homonuclear correlation spectroscopy (^1^H−^1^H COSY, bold lines) and heteronuclear multiple bond correlation (HMBC, arrows) correlations.

**Figure 2 molecules-25-02077-f002:**
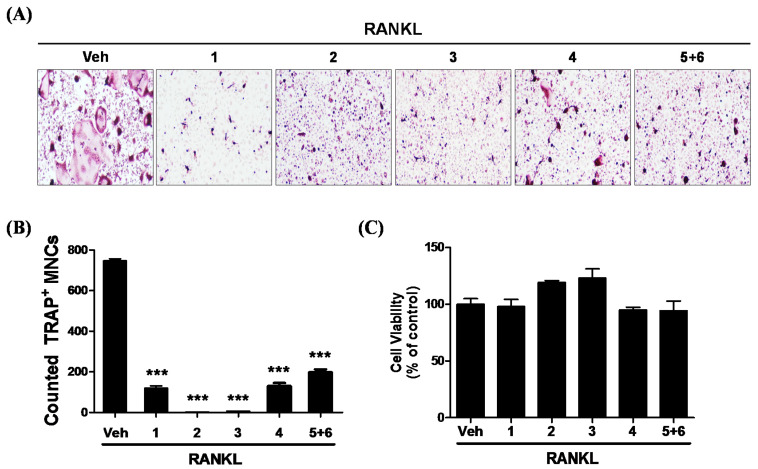
The isolated compounds inhibit osteoclast formation in bone marrow derived macrophages (BMDMs). (**A**) BMDMs were pretreated with or without compounds **1**–**4** (50 μM) or **5**+**6** (40 μM) in the presence of macrophage colony-stimulating factor (M-CSF) (25 ng/mL) for 2 h and subsequently stimulated with receptor activator of nuclear factor-κB (RANKL, 100 ng/mL) for 4 days. Multinucleated cells were visualized by tartrate-resistant acid phosphatase (TRAP) staining. (**B**) TRAP-positive multinucleated cells were counted to determine osteoclast numbers. Data are presented as means ± SD (*n* = 3). *** *p* < 0.001 compared with the Vehicle (Veh). (**C**) BMDMs were cultured on a 96-well plate and treated with the compounds for 24 h in the presence of M-CSF. Cell viability was determined as described in Materials and Methods. Data are presented as means ± SD (*n* = 3).

**Figure 3 molecules-25-02077-f003:**
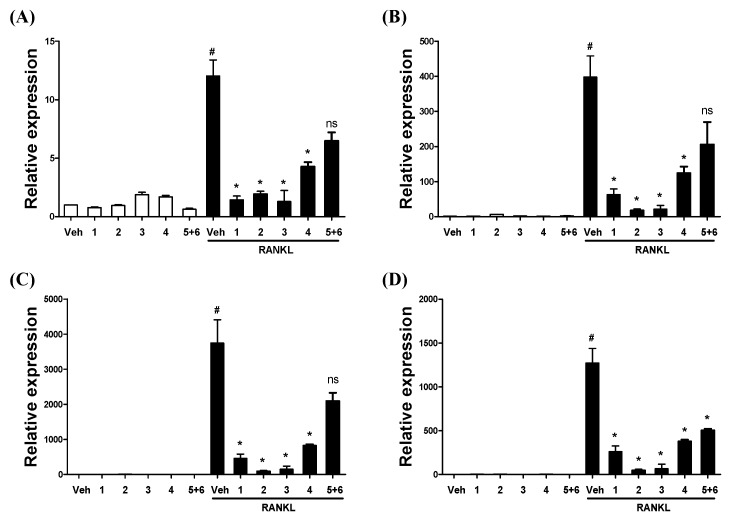
The isolated compounds regulate RANKL-mediated osteoclast-specific gene expression. (**A**–**D**) BMDMs were pretreated with or without **1**–**4** (50 μM) or **5**+**6** (40 μM) in the presence of M-CSF (25 ng/mL) for 2 h and subsequently stimulated with RANKL (100 ng/mL) for 4 days. Their total RNA was prepared, followed by RT-PCR using the indicated primers for (**A**) NFATc1, (**B**) TRAP, (**C**) Cathepsin K, and (**D**) DC-STAMP. Data are presented as means ± SD (*n* = 3). ^#^
*p* < 0.05 compared with the Vehicle (Veh), * *p* < 0.05 compared with the RANKL group. ns: not significant.

**Figure 4 molecules-25-02077-f004:**
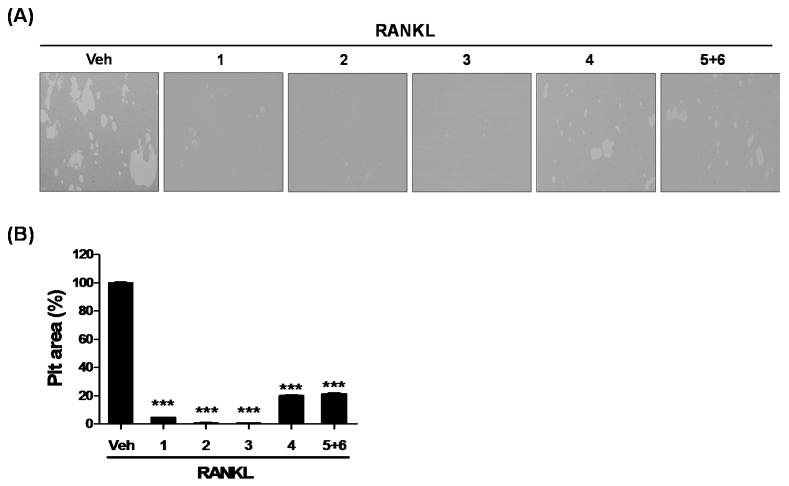
The isolated compounds inhibit RANKL-induced bone resorption function. (**A**) BMDMs were pretreated with or without **1**–**4** (50 μM) or **5**+**6** (40 μM) in the presence of M-CSF (25 ng/mL) for 2 h on an Osteo assay plate and subsequently stimulated with RANKL (100 ng/mL) for 4 days. (**B**) Pit areas were quantified using Image J software. ***, *p* < 0.001 compared with the Vehicle (Veh).

**Figure 5 molecules-25-02077-f005:**
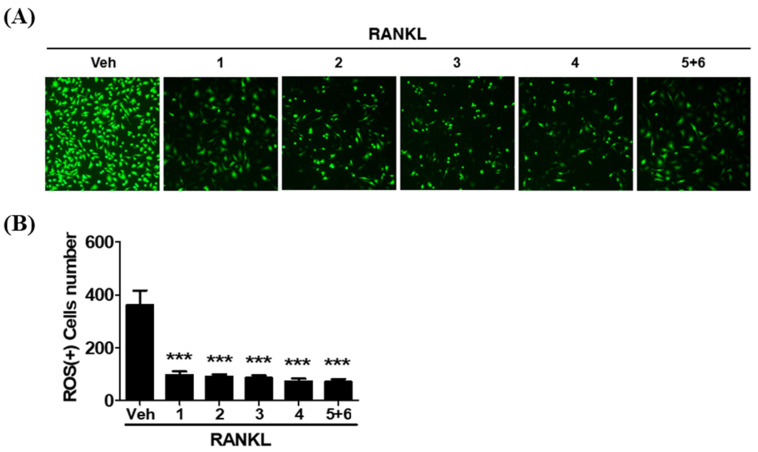
The isolated compounds attenuate RANKL-induced intracellular reactive oxygen species (ROS) production during osteoclast differentiation. (**A**) BMDMs were pretreated with or without **1**–**4** (50 μM) or **5**+**6** (40 μM) in the presence of M-CSF (25 ng/mL) for 2 h and subsequently stimulated with RANKL (100 ng/mL) for 2 days. Cells were incubated with 15 μM 2′, 7′-dichlorofluorescin diacetate (DCFDA) for 20 min and observed under fluorescence microscopy. (**B**) Quantification of ROS-positive cells number in each well. *** *p* < 0.001 compared with the Vehicle (Veh).

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
