# Peer review of "New 8-C-p-Hydroxylbenzylflavonol Glycosides from Pumpkin (Cucurbita moschata Duch.) Tendril and Their Osteoclast Differentiation Inhibitory Activities"

_molecules, 2020, doi:10.3390/molecules25092077_

Round 1
Reviewer 1 Report
To the Authors,
The authors purified novel compounds from Pumpkin Tendril and investigated the inhibitory effect of these compounds on the osteoclast differentiation in mouse bone marrow-derived macrophages by real-time quantitative PCR assay. Although Pumpkin Tendril might promote beneficial effects on the osteoclast differentiation in vitro, the results in this study do not have a clinical impact on human health science since Pumpkin Tendril, but not Pumpkin Fruits, is not generally eaten as food. The cell toxicity of these compounds from their experiments is also questionable. Furthermore, the mechanisms of the inhibitory effect of these compounds on osteoclast differentiation have not been elucidated in the manuscript. Therefore, I believe that the manuscript does not have a substantial impact on this field.
Author Response
The authors purified novel compounds from Pumpkin Tendril and investigated the inhibitory effect of these compounds on the osteoclast differentiation in mouse bone marrow-derived macrophages by real-time quantitative PCR assay. Although Pumpkin Tendril might promote beneficial effects on the osteoclast differentiation in vitro, the results in this study do not have a clinical impact on human health science since Pumpkin Tendril, but not Pumpkin Fruits, is not generally eaten as food. The cell toxicity of these compounds from their experiments is also questionable. Furthermore, the mechanisms of the inhibitory effect of these compounds on osteoclast differentiation have not been elucidated in the manuscript. Therefore, I believe that the manuscript does not have a substantial impact on this field.
<Answer> We appreciate your kind advice. Pumpkin stems with leaves, shoots, and tendril are usually eaten as vegetables after blanching in some Asian countries, especially Korea. In particular, pumpkin tendril has been used as a folk remedy to prevent bleeding and to reduce abdominal pain in pregnant women and to attenuate menopause-related symptoms, and its effects are documented in “Donguibogam’, a famous oriental medicine book of Korea. Additionally, we have demonstrated that pumpkin tendril suppresses osteoclastogenesis by down-regulating p38 and ERK signaling RANKL-treated BMDMs. Therefore, we believe that pumpkin tendrils can prevent or delay osteoporosis via suppression of osteoclastogenesis in clinical study. In addition, the chemical constituents identified in this study may provide valuable information for evaluating health-beneficial effects of this plant as well as utilizing marker compounds in its-processed foods.
The cell toxicities of the isolated compounds were already represented in in Figure 2C. To determine the suitable concentration of new compounds on osteoclast differentiation, cell viability was assessed with 1, representatively. BMDMs were pretreated with various concentrations (12.5, 25, 50, 75 μM) of 1 in the presence of M-CSF for 24 h and cell viability was determined by MTT. Compound 1 showed significant cytotoxicity at concentration of 75 μM, whereas there was no cytotoxicity under 50 μM (Supplementary Figure 47). However, a concentration of 40 μM was used for the 5+6 mixture, as cytotoxicity was observed at a concentration of 50 μM during osteoclast differentiation (Supplementary Figure 48A and 48B). Therefore, we determined the concentrations of 50 μM (compounds 1–4) and 40 μM (compounds 5+6) to evaluate the inhibitory effect of RANKL-induced osteoclast differentiation.
These findings suggest that six new 8-C-p-hydroxybenzylflavonol glycosides (1–6), especially compounds 2 and 3, isolated from the hot water extract of pumpkin tendril inhibit osteoclast differentiation by decreasing the expression of genes related to osteoclastogenesis. Furthermore, we confirmed that bone resorption was inhibited by isolated compounds. Recently, we reported that water extract of pumpkin tendril suppresses RANKL-induced osteoclast differentiation and bone resorption by suppressing RANKL-mediated p38 and ERK phosphorylation [17]. In addition, water extract of pumpkin tendril treatment decreased the expression of osteoclast-related genes such as NFATc1, TRAP, Cathepsin K and TRAP [17]. In our study, new compounds isolated from water extract of pumpkin tendril showed different levels of inhibitory effect at the same concentration. This means that the inhibitory effect of pumpkin tendril extract on osteoclastogenesis acted through a complex mechanism of several compounds. These results and discussions were described in in lines 225-244 of revised manuscript. We would be very grateful if you understand this.
<Reference 17> Choi, J.H.; Jang, A.R.; Jeong, H.N.; Kim, K.K.; Kim, Y.M.; Cho, J.Y.; Park, J.H. Water extract of tendril of Cucurbita Moschata Duch. suppresses RANKL-induced osteoclastogenesis by down-regulating p38 and ERK signaling. Int. J. Med. Sci. 2020, 17, 632-639.
Reviewer 2 Report
In this paper, Kiok Kim et al. showed how the 8-C-p-hydroxybenzylflavonol glycosides extracted from pumpkin tendril (Cucurbita moschata Duch.) were able to inhibit the osteoclasts differentiation. They performed a careful characterization of the chemical compounds and targeted experiments to demonstrated their potential role in osteoclast biology. Overall, the paper was well constructed and easy to follow for the reader.
In the introduction (line 40-43) Cappariello et al,2014 (Biotechnological approach for systemic delivery of membrane Receptor Activator of NF-κB Ligand (RANKL) https://doi.org/10.1016/j.biomaterials.2014.12.033) active domain into the circulation) should be cited by the authors. Please quote the paper.
In the results, the authors showed that some of the compounds derived from Pumpkin Tendril were able to reduce dramatically the osteoclast differentiation from BMDMs. It could be interesting to see the effect on the osteoclast function. For this purpose, the author should plate differentiated osteoclasts on a bovine bone slice and treat them with their compounds and evaluated the bone resorption area (or the pit index) after 3 days. Alternatively, they can treat the differentiated osteoclasts for 3 days, and then, evaluate the osteoclast maturation by counting the number of nuclei.
Please, review the English in the introduction.
Author Response
In this paper, Kiok Kim et al. showed how the 8-C-p-hydroxybenzylflavonol glycosides extracted from pumpkin tendril (Cucurbita moschata Duch.) were able to inhibit the osteoclasts differentiation. They performed a careful characterization of the chemical compounds and targeted experiments to demonstrated their potential role in osteoclast biology. Overall, the paper was well constructed and easy to follow for the reader.
1) In the introduction (line 40-43) Cappariello et al,2014 (Biotechnological approach for systemic delivery of membrane Receptor Activator of NF-κB Ligand (RANKL) https://doi.org/10.1016/j.biomaterials.2014.12.033) active domain into the circulation) should be cited by the authors. Please quote the paper.
<Answer> We appreciate your kind comment. We cited this according to your suggestion. References were re-arranged in our revised manuscript.
2) In the results, the authors showed that some of the compounds derived from Pumpkin Tendril were able to reduce dramatically the osteoclast differentiation from BMDMs. It could be interesting to see the effect on the osteoclast function. For this purpose, the author should plate differentiated osteoclasts on a bovine bone slice and treat them with their compounds and evaluated the bone resorption area (or the pit index) after 3 days.
<Answer> We believe that your suggestions and comments are helpful to improve our study. As you suggested, we performed additional experiment to the bone resorption assay using Osteo assay surface plate. We added the information in Experimental Section (3.5.4). Results were shown below and added in the manuscript as Figure 4.
→ Mature osteoclast caused the formation of pits in the presence of M-CSF and RANKL. Therefore, we investigated whether each compound has the potential to inhibit the function of osteoclastic bone resorption. In order to examine the effect of isolated compounds on osteoclastic bone resorption, BMDMs were cultured on an Osteo assay surface plate and pretreated with 1–4 (50 μM) or 5+6 (40 μM) in the presence of M-CSF (25 ng/mL) for 2 h, and subsequently stimulated with RANKL (100 ng/mL) for 4 days. Consistent with the size and number of osteoclasts (Figure 2A and 2B), the isolated compounds reduced remarkably the formation of bone resorption by inhibiting the area and number of pits on the Osteo surface (Figure 4A and 4B). These findings suggest that six new 8-C-p-hydroxybenzylflavonol glycosides (1–6) compounds, especially compound 2 and 3, isolated from the hot water extract of pumpkin tendril inhibit osteoclast differentiation by decreasing the expression of genes related to osteoclastogenesis. Furthermore, we confirmed that bone resorption was inhibited by isolated compounds. However, the detailed mechanism by which compound 5+6 suppresses RANKL-induced osteoclast differentiation and bone resorption remains to be elucidated.
3) Alternatively, they can treat the differentiated osteoclasts for 3 days, and then, evaluate the osteoclast maturation by counting the number of nuclei.
<Answer> The mature osteoclasts are giant cells with three or more nuclei and highly express TRAP enzyme. In our study, osteoclast was stained with the osteoclast marker TRAP, and osteoclast differentiation was evaluated by counting TRAP-positive multinucleated cells (TRAP+MNCs). Please check Figure 2B.

Reviewer 3 Report
Kim et al. demonstrated the osteoclast differentiation inhibitory potential of six newly characterized flavonol glycosides from C. moschata in a well planned and executed in vitro experimental model of BMDM and presented the obtained results as well as the discussion in clear and concise manner. However, the following minor corrections are suggested to be addressed for better clarity and to improve the overall quality of the paper;
- Provide rationale for the doses (50 µM for 1-4 and 40 µM for 5+6) used in the study and why was 5 and 6 added for the treatment and not used as 1 - 4? This must be justified and included in the manuscript.
- Please, give explanation for the non-significant effect of 5+6 treatment on the gene expression biomarkers except for DC-STAMP?
- Kindly delete the last statement of section 3.1 or what is this supposed to mean? (Lines 220 -221).
- Provide suitable reference for the cell viability assay (section 3.5.2) and include a statement on the confluency of the cells.
Author Response
Kim et al. demonstrated the osteoclast differentiation inhibitory potential of six newly characterized flavonol glycosides from C. moschata. in a well planned and executed in vitro experimental model of BMDM and presented the obtained results as well as the discussion in clear and concise manner. However, the following minor corrections are suggested to be addressed for better clarity and to improve the overall quality of the paper
- Provide rationale for the doses (50 µM for 1-4 and 40 µM for 5+6) used in the study and why was 5 and 6 added for the treatment and not used as 1 - 4? This must be justified and included in the manuscript.
<Answer> In this study, compounds 5 and 6 were purified as mixture from pumpkin tentril. The 5 and 6 mixture were determined to be 8-C-p-hydroxybenzylisorhamnetin 3-O-rutinoside [8-C-p-hydroxybenzylisorhamnetin 3-O-(α-L-rhamnopyranosyl(1→6)-β-D-glucopyranoside] and 8-C-p-hydroxybenzylisorhamnetin 3-O-(α-L-rhamnopyranosyl(1→6)-β-D-galactopyranoside, based on the MS and NMR experiments. We attempted to separate two compounds by chromatographic tools to evaluate their inhibitory effects against the osteoclast differentiation. However, it was difficult to separate two compounds, respectively, because they were stereoisomers that have the same molecular formula. We thought that the aglycone (8-C-p-hydroxybenzylisorhamnetin) rather than the sugar moieties in two compounds contributes to the inhibition of the osteoclast differentiation. Therefore, to determine the inhibitory effect of RANKL-induced osteoclast differentiation, 5 and 6 were assayed in the mixture. Such reason was described in in lines 179-180 of revised manuscript.
- Please, give explanation for the non-significant effect of 5+6 treatment on the gene expression biomarkers except for DC-STAMP?
<Answer> Thank you for your kind comment. We added the explanation in the Results & Discussion.
→ Upon binding to RANK on the surface of osteoclast precursor, RANKL induces the expression of NFATc1, a master regulator of osteoclastogenesis that regulates osteoclast-related genes such as TRAP, Cathepsin K, and DC-STAMP [20-22]. TRAP and Cathepsin K are involved in the degradation of bone. DC-STAMP is required for cell to cell fusion for the generation of intact multinucleated osteoclasts. Thus, we examined whether the isolated compounds inhibited RANKL-induced gene expression. --------- Treatment of Compound 5+6 trended towards inhibiting the expression of the RANKL-induced NFATc1, TRAP and Cathepsin K mRNA, although this was not statistically significant (Figure 3A, 3B, and 3C). However, the expression of the DC-STAMP mRNA was effectively attenuated by all of the tested compounds, including 5+6 (Figure 3D). It can be seen that Compound 5+6 reduced the size and proliferation of mature osteoclasts as a result of Figure 2A and B by suppressing DC-STAMP, which plays a role of cell fusion.
- Kindly delete the last statement of section 3.1 or what is this supposed to mean? (Lines 220 -221).
<Answer> This sentence was deleted.
- Provide suitable reference for the cell viability assay (section 3.5.2) and include a statement on the confluency of the cells.
<Answer> We added the reference in the Experimental Section (section 3.5.2).
Round 2
Reviewer 1 Report
To The Authors,
The authors revised all of the issues of their manuscript, and the manuscript is suitable for the publication of the Molecules.